# WCD Ideas: Teleconnections through weather rather than stationary waves

Clemens Spensberger[1]

[1]Geophysical Institute, University of Bergen, and Bjerknes Centre for Climate Research, Bergen, Norway

**Correspondence:** Clemens Spensberger (clemens.spensberger@uib.no)

**Abstract.** Conventionally, teleconnections in the atmosphere are described by correlations between monthly mean fields. These correlations are supposedly caused by stationary Rossby waves. The main hypothesis explored in this idea is that teleconnections are instead established by chains of events on synoptic time scales, that is by weather. Instead I hypothesise that non-stationary Rossby waves play an important role in establishing teleconnections. If these hypotheses are correct, much of the vast literature on this topic misses an essential part of the atmospheric dynamics leading to teleconnections.

## 1 Introduction

Conventionally, teleconnections are described by statistical relations between time-mean fields. For example, many teleconnections are defined through EOF analyses based on monthly mean sea-level pressure or geopotential (e.g., Wallace and Gutzler, 1981; Thompson and Wallace, 2000). Thus defined, teleconnections statistically describe spatial relations in how these fields vary.

Teleconnections can be comparatively confined regionally. An example would be the North Atlantic Oscillation (NAO), an anticorrelation between the sea-level pressure over Iceland and the Azores. I call these teleconnections "regional" because their spatial scale is comparable to that of a single weather system. For example, the anticorrelation defining the NAO can be physically understood as variations in the occurrence of a characteristic weather event (Rossby wave-breaking) over the North Atlantic (Woollings et al., 2008).

Other teleconnections extend over much larger distances. For example, variations in tropical convection in the Indo-Pacific associated with the Madden-Julian Oscillation (MJO) can influence weather over the North Atlantic despite the large distance between these regions (e.g., Cassou, 2008; Garfinkel et al., 2014; Fromang and Rivière, 2020). It is these long-distance teleconnections that are the focus of this idea. They cannot be explained by variations in a single weather event, such that other processes must establish the observed connection between the distant regions.

In a time-mean perspective, such long-distance teleconnections are often associated with a wave pattern, which, following the pioneering work of Hoskins and Karoly (1981) and a somewhat more recent conceptual review by Held et al. (2002), is generally interpreted as a stationary Rossby wave. The stationary wave paradigm has since dominated the teleconnection

literature[1]. For reasons that will become clear in the following, I am challenging the prevailing paradigm and propose new
frameworks to analyse and interpret teleconnections.

Returning first to the example of teleconnections between the North Atlantic and the MJO, several teleconnection pathways have been documented to contribute to this connection (Liu and Alexander, 2007; Stan et al., 2017). The MJO is thought to affect the North Atlantic via stationary waves interacting with the North Pacific storm track, which in turn has a downstream influence on the North Atlantic (e.g., Stan et al., 2017). In addition, both oscillations are thought to affect the stratospheric
polar vortex, which can subsequently exert a downward influence on the North Atlantic storm track (Jiang et al., 2017). Finally, Rossby waves emanating from the North Atlantic storm track can trigger MJO initiation events in the tropical Indian Ocean (e.g., Lin et al., 2009).

Several aspects of these teleconnection pathways have been associated with variations in the occurrence of weather events (schematic overview for the MJO in Fig. 1 of Stan et al., 2017). The MJO has been associated with variations in the occurrence
of both blocking and atmospheric river events in the eastern North Pacific (e.g., Moore et al., 2010; Payne and Magnusdottir, 2014). For the El-Niño Southern Oscillation (ENSO), Schemm et al. (2018) documented a relation to the prevailing locations of cyclogenesis over North America and along the North American east coast, but their results have not been translated to MJO-forcing yet. While these associations have enriched the interpretation of these teleconnections, they have not changed the prevailing dynamical interpretation as stationary waves (e.g., Stan et al., 2017; Fromang and Rivière, 2020).
Understanding these teleconnections is crucial for subseasonal-to-seasonal (s2s) prediction of the North Atlantic storm track. In isolation and deterministically, the mid-latitude troposphere is only predictable for about two weeks (e.g., Lorenz, 1969, 1982, 1996; Zhang et al., 2007, 2019; Domeisen et al., 2018). The current practical limit of predictability is even well below this theoretical limit (e.g., Zhang et al., 2019; Selz et al., 2022), although ensemble forecasting helps increase the limit of deterministic predictability to some extent (e.g., Leutbecher and Palmer, 2008). As a predictable forcing of the storm track,
the MJO is one of the likely sources of s2s predictability of the North Atlantic (e.g., Scaife et al., 2014, 2017). Some practical predictability of the North Atlantic storm track has been documented on s2s scales (e.g., Palmer et al., 2004; Scaife et al., 2014; Vitart, 2017; Stan et al., 2022).

Despite the widespread use of the Hoskins and Karoly (1981) arguments to explain long-distance teleconnections, this perspective has severe limitations in explanatory power, which so far have remained largely unaddressed. The Hoskins and
Karoly (1981) arguments are based on a time-mean perspective and require the definition of a basic state on which small-amplitude perturbations propagate linearly. Given these strong assumptions, linear stationary wave theory following Hoskins and Karoly (1981) is surprisingly successful in describing observed time-mean states (Held et al., 2002; Potter et al., 2013). Despite this success, however, this description remains only self-consistent in that it cannot explain how the time-mean state with apparently linear wave perturbations can emerge from chaotic, non-linear weather. There is no obvious link from a time-
mean state back to the instantaneous weather from which it emerged.

On the contrary, inferences about wave propagation based on a time-mean state can be misleading. As Potter et al. (2013) demonstrated, small changes in the background state can be enough to cause large changes in wave reflection, and thus the

---

[1]On 29 January 2024 Google Scholar listed 3159 citations of Hoskins and Karoly (1981), of which 791 in the five years 2019-2023.

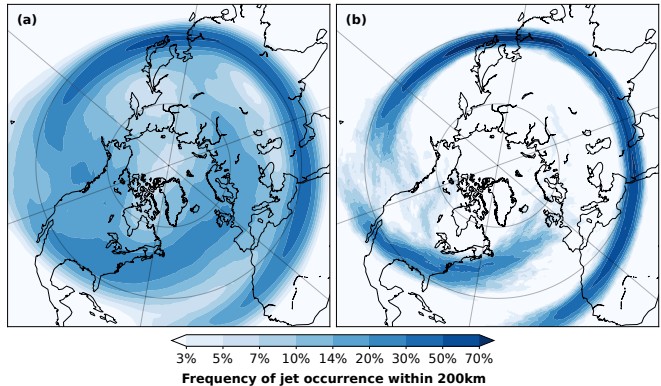

**Figure 1.** The effect of monthly averaging for detected jet axes. The detections are based on (a) 3-hourly instantaneous winds, or (b) monthly mean winds in ERA5 during winter for the period 1979-2022. Jets in (b) deviate little from their climatological mean position. Jets are detected following Spensberger et al. (2017).

direction of wave propagation. As illustrated by Fig. 1, there are typically large differences between instantaneous and time-mean flow. In monthly means, jets deviate hardly at all from their climatological position. On shorter time scales, however, jets

occur over a wide range of latitudes. Rossby waves tend to propagate in the direction of the jet (e.g., Martius et al., 2010; White et al., 2022), such that it seems highly unlikely that the instantaneous wave propagation can be inferred from a time-mean state, and thus that stationary wave theory can provide a causal explanation for the emergence of long-distance teleconnections.

## 2  Hypotheses

The missing causal explanation calls for a reinterpretation of long-distance teleconnections in terms of weather. As the basis

for this reinterpretation, I hypothesise

1. Long-distance teleconnections are established through (chains of) events on synoptic time scales, i.e., "weather".

2. These chains of events are orchestrated by predominantly non-stationary Rossby waves.

3. Predictable forcing is communicated through long-distance teleconnections by weather events.

While the scope of these hypotheses is global, a first step could be focusing on teleconnections between the MJO and the North

Atlantic. These specific teleconnections are an ideal testbed because (a) the North Atlantic storm track is very well studied, and (b) significant but weak correlations with the MJO suggest large variability within the teleconnections.

Analogous hypotheses have been shown to be correct for air-sea and air-ice interactions. Here, the interpretation of (monthly) mean fluxes can be quite misleading (Ogawa and Spengler, 2019) because brief bursts dominate the climatological exchange and its variability in the mid-latitudes. For example, Greenland tip jets shape deep ocean convection in the Irminger Sea despite

being short-lived and relatively small-scale (e.g., Pickart et al., 2003; Piron et al., 2016). Similar bursts in the air-sea exchange

are associated with extratropical cyclones (Sorteberg and Kvingedal, 2006; Sampe and Xie, 2007), polar lows (Condron and Renfrew, 2013) and cold-air outbreaks (Papritz and Spengler, 2017; Aemisegger and Papritz, 2018). If weather events dominate the climatology of these exchange processes, it seems plausible that weather events also dominate the exchange, for example, between the tropics and extratropics, or within long-distance teleconnections. If the analogy holds, one might conceptualise the teleconnection between the MJO and the North Atlantic as an intermittently occurring event in a similar way as Aemisegger and Papritz (2018) conceptualise air-sea exchange through an event perspective.

Further, Davies (2015), Röthlisberger et al. (2019), and Ali et al. (2021) use an analogous approach when discussing the relation between recurring Rossby wave packets and extreme events, months, and seasons. In line with my hypothesis 2, these authors point out the role of non-stationary Rossby waves in shaping both the evolution of weather on synoptic time scales as well as the respective time mean states. They thus make plausible hypothesis 2 for the subset of time average states that are dominated by recurring Rossby waves. Conversely, hypothesis 2 constitutes a generalisation of their approach to time-mean states in general where the link between the time-mean and instantaneous patterns is less clear-cut.

While providing an alternative perspective on teleconnections, hypotheses 1-3 provide no guidance on why the dominant stationary-wave perspective on these teleconnections appears so immensely successful (cf., citations of Hoskins and Karoly, 1981; Held et al., 2002). Stationary Rossby waves appear ubiquitously in monthly, composite, and climatological means, and have with some success been applied to study s2s predictability of the North Atlantic (e.g., Scaife et al., 2017).

To clarify the relation between the weather-based and stationary wave-based perspectives on teleconnections, it is useful to consider an analogy between the time-mean perspective and geostrophy. In both geostrophic balance and time-mean perspective, information about causality is lost. Neither does the geostrophic wind cause the pressure gradient, nor vice-versa. Analogously, there is no causal relation from a time-mean state back to the instantaneous events from which it emerged. Further, causal relations in the evolution of weather do not translate to, for example, the succession of monthly averages. I therefore hypothesise

4. Time-mean wave patterns are a symptom rather than a cause of long-instance teleconnections.[2]

The analogy also reveals wide gaps in our understanding. While geostrophy is well-founded (a) on scaling arguments that explain why the balance exists and (b) on geostrophic adjustment theory that explains how geostrophic balance can be attained in practice, neither of these ingredients exists for the link between the instantaneous and time-mean perspective on weather. Neither do we know why we should expect time mean states to apparently follow linear wave theory, nor do we know how chaotic, non-linear weather can reduce to an time-mean states that apparently follow linear theory.

If correct, hypothesis 2 provides the missing conceptual foundation. Non-stationary finite-amplitude Rossby waves regularly propagate approximately linearly over large distances (e.g., Wirth and Eichhorn, 2014; O'Brien and Reeder, 2018). They further have a clear influence on the non-linear evolution of mid-latitude weather, for example by determining the predominant locations of cyclogenesis (Holton and Hakim, 2013). It thus seems plausible that non-stationary Rossby waves constitute the ordering principle that links the non-linear instantaneous weather to mean states that appear to follow linear stationary theory.

---

[2]For a more in-depth discussion of the issue of causality in this context please refer to the open discussion around comment 2 of reviewer 1. The comment is available under https://doi.org/10.5194/egusphere-2023-2353-RC1, the response under https://doi.org/10.5194/egusphere-2023-2353-AC1.

## 3 Conceptual risks and potential impact

These hypotheses challenge the dominant paradigm of how long-distance teleconnections arise. This paradigm has been prevailing since the pioneering work of Hoskins and Karoly (1981) and has generally successfully been applied to link zonal asymmetries and mid-latitude variability to faraway orography and diabatic forcing (e.g., Held et al., 2002; Scaife et al., 2017). There is thus a considerable conceptual risk that the hypotheses will need to be refuted despite my above arguments.

If, however, the hypotheses turn out to be true, we need to conceptually reframe how long-distance teleconnections are established in the atmosphere, shifting the focus from monthly and longer time scales to synoptic time scales. This reframing is synonymous with a deeper physical understanding because it transforms teleconnections from statistical relations to a causal chain of events, in which each link in the chain depends on well-defined conditions. In this respect, the approach I suggest to conceptualise teleconnections is similar in spirit to the Shepherd et al. (2018) storylines approach to represent uncertainty around singular events.

The deeper understanding of teleconnections provides the basis for a better understanding of the potential and limits for predictability through these teleconnections. In a time-mean perspective, for example, stationary wave patterns connecting the MJO with the North Atlantic can only be analysed as a whole, whereas the weather perspective allows one to follow the causal chain of events link by link. This is advantageous because for every link in isolation it is much easier to physically understand the conditions under which it is effective than for the wave pattern as a whole.

This perspective on the conditions for predictability will also help unravel the so-called predictability paradox (Scaife et al., 2014; Scaife and Smith, 2018). The paradox is rooted in some s2s ensemble predictions systems being too dispersive, leading to the rather paradoxical result that members in such ensembles are better at predicting reality than each other (Scaife and Smith, 2018). If the hypotheses are correct, this paradox implies that at least one of the links in the chain of events that constitutes the teleconnection is simulated as much more uncertain than this link actually is. The suggested conceptual reframing of teleconnections breaks down the assumed information transfer by stationary waves into several synoptic-scale constituents which can be analysed in isolation. In other words: if the simulated stationary wave pattern deviates from the observed, there are myriad potential causes of the issue in a numerical weather prediction model. If, in contrast, the modification of propagating Rossby wave packets was simulated to be systematically more uncertain than it appears in observations over a cyclogenesis region, the list of potential processes causing the issue has become much shorter. The case study of González-Alemán et al. (2022) illustrates how such an effort might look in practice, showing that the tropospheric predictability following the 2018 sudden stratospheric warming event was mediated by two cyclogenesis events.

Finally, even if the hypotheses will need to be refuted, efforts to systematically test them would have a considerable scientific impact. The approach envisioned to test the hypotheses entails the compilation of a comprehensive dataset showing both stationary and non-stationary Rossby wave activity during the past decades. This dataset provides a new avenue to address long-standing issues on the relation between the near-stationary and transient circulation, as well as the relation between waves and weather. By conceptually linking waves and weather, this avenue also constitutes a bridge between the synoptic and weather event-based perspective prevalent in dynamical meteorology and the eddy-mean flow perspective prevalent in

climate dynamics. Finally, the dataset of Rossby wave activity will be valuable to clarify the link between Rossby waves and extreme events suggested by many case studies of (in particular) flood events (e.g., Massacand et al., 1998; Enomoto et al., 2007; Martius et al., 2008; Wirth and Eichhorn, 2014; Röthlisberger et al., 2016).

## 4 Review of diagnostics for Rossby wave activity

As Rossby waves have long been recognised as a key feature of the dynamics of the atmosphere in the mid-latitudes, many diagnostics have been developed to capture their effect on all time scales from synoptic to climatological. One prominent branch of such diagnostics aims to quantify the effect of transients in general, or Rossby waves in particular, on a basic state. This branch of diagnostics starts with the Eliassen-Palm flux (Eliassen, 1961), the divergence of which diagnoses the effect of stationary waves on the time and zonal mean circulation. But while the Eliassen-Palm flux is a powerful tool to understand the time-mean circulation of the atmosphere, it cannot quantify the effect of individual wave packets. As a step in this direction, Hoskins (1983) and Trenberth (1986) generalised the Eliassen-Palm flux to not require a zonal average, and to require a time average over one wave period only. Similarly, the Plumb (1986) flux only requires an average over a wave period and provides momentum fluxes due to transients in both the horizontal and vertical.

Despite these generalisations, these diagnostics cannot be applied to snapshots of the atmosphere, because they still require some temporal averaging. Plumb (1985) was the first to derive a wave-activity flux in which the wave activity could be diagnosed without averaging over one wave period or wavelength. He however required a zonal average basic state for his derivation. His derivation was later extended to other specific basic states, but the applicability of his wave-activity flux remained somewhat limited. Finally, Takaya and Nakamura (2001) and Wolf and Wirth (2015) generalised the Plumb (1985) flux to general basic states by assuming that the wave perturbations are quasi-geostrophic or semi-geostrophic in nature, respectively.

Both Takaya and Nakamura (2001) and Wolf and Wirth (2015) assume small-amplitude perturbations. Nakamura and Zhu (2010), Nakamura and Solomon (2010, 2011), and Methven and Berrisford (2015) successfully lifted this restriction by rebasing the wave-activity flux formalism on finite-amplitude departures of a potential vorticity contour from its equivalent latitude. These equivalent latitudes represent the basic state and are defined through a so-called modified Lagrangian mean state. While this framework is conceptually appealing, the required mean state cannot represent zonal asymmetries and is difficult to define in practice (e.g., Methven, 2013; Methven and Berrisford, 2015). Teubler and Riemer (2016) follow a more pragmatic, but mathematically less stringent approach by considering potential vorticity departures from a rolling 30-day average. Similar in spirit, Polster and Wirth (2023) define zonally asymmetric basic states by defining equivalent latitudes in by a rolling 60° longitude window.

While the aforementioned diagnostics are explicitly designed to capture finite-amplitude wave activity, they still depend on a basic state. This dependence on a basic state is also the main caveat of alternative diagnostics developed to capture the effect of Rossby waves. For example, Orlanski and Katzfey (1991) propose to use eddy kinetic energy fluxes to this end. Their

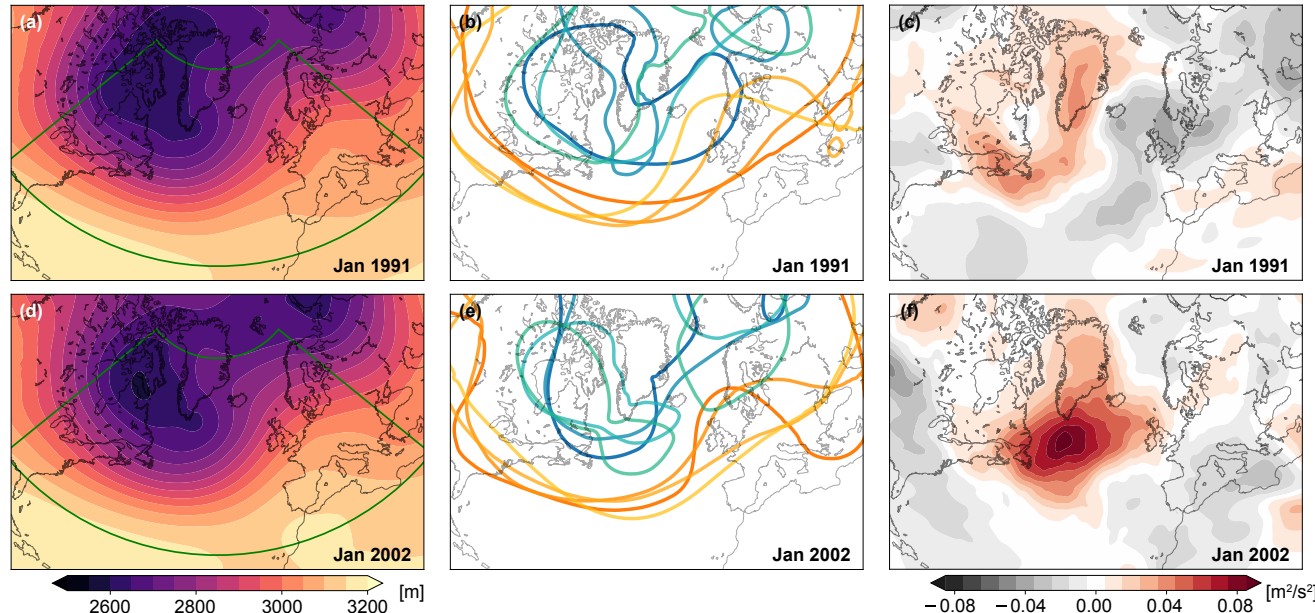

**Figure 2.** Weak relation between monthly means and weekly means. Panels (a, d) show the two Januaries with the most similar mean 700-hPa geopotential distribution within the ERA-Interim dataset for 1979-2018 within the green box. The spaghetti plots (b, e) show weekly averages of 700 hPa geopotential of the first four weeks of each January with contours at 2600 m (blue to cyan, in sequence) and 3000 m (orange to yellow). Although, for example, week 1 of January 1991 is similar to week 4 of 2002, in general the weekly averages are different enough within each month and across the two Januaries to imply considerably different wave propagation. This is emphasized in panels (c, f) by the mean forcing of the near-stationary streamfunction (time scales ≥ 30 days) by the transient circulation (time scales < 30 days). This forcing differs widely between these two months, implying different chains of weather events and a different evolution of the near-stationary circulation despite the similarity in monthly mean state.

diagnostic is in principle applicable to generic basic states; in practice they suggest using a 30-day average centred on the synoptic event in question.

Any such decomposition into a basic state and perturbation poses a severe conceptual challenge, similar in spirit to the limitations of the stationary wave perspective on teleconnections. Defining a basic state necessarily constitutes a trade-off between, on the one hand, being specific enough to represent the instantaneous wave propagation, and, on the other hand, being smooth enough for the diagnostics to remain applicable. The shortest-term averages in the aforementioned diagnostics are the 30-day rolling averages used by Orlanski and Katzfey (1991) and Teubler and Riemer (2016). As illustrated by Fig. 2, even for such comparatively short-term averages, the mean state (Fig. 2a, d) remains a poor representation of the varying conditions non-stationary Rossby waves might encounter, here exemplified by weekly averages (Fig. 2b, e). The weekly averages in Fig. 2b, e not only deviate considerably from the respective monthly average, but they also differ considerably between months with very similar average states. This is consequential, because for the months shown in Fig. 2, all diagnostics discussed

so far would infer nearly identical wave propagation characteristics, while the actual conditions differed considerably. As a reference, a week is long enough for a Rossby wave packet to propagate from the central North Pacific via North America to Europe (where it led to severe flooding; Enomoto et al., 2007). For such fast-propagating wave packets even the weekly time scale might be too long to represent the actual propagation characteristics.

Panels (c, f) further illustrate that a similar monthly average can result from different weather patterns. The panels show the mean streamfunction tendency of the near-stationary circulation (time scales $\geq 30$ days) as forced by the transient circulation, following in spirit the derivations in Cai and Van Den Dool (1994) and Feldstein (1998). With time scales of less than 30 days, the transient circulation here consists of weather and the sub-monthly variability of its patterns. The forcing by the transients of the near-stationary circulation is markedly different between the two months (Fig. 2c, f), demonstrating that a similar state of the near-stationary circulation does neither determine the transient weather patterns, nor the direction into which the near-stationary circulation is evolving.

Because of these conceptual challenges, it would be highly desirable to diagnose wave activity without requiring a decomposition into basic state and transients. One way to accomplish this would be to define and identify Rossby wave packets through the automated feature detection algorithm in Wirth et al. (2018), but this diagnostic comes with a new set of assumptions. In particular, both the region and direction of propagation must be defined a-priori, which considerably limits the applicability of this method.

## 5 Alternative approach: idealised reanalyses

The conceptual limitations of separating the atmospheric state into mean and perturbation have been noted and discussed many times before (e.g., Held et al., 2002; Potter et al., 2013; Wirth and Polster, 2021; White et al., 2022); these limitations are also often mentioned in the studies reviewed in the previous section. The continued reliance on such a separation, however defined, might thus be more out of the lack of a better alternative than out of conviction. The following suggestion of an alternative approach to diagnose Rossby wave activity does not require a separation into basic state and perturbation.

The core of the alternative approach is a new kind of dataset which I call *idealised reanalysis*. In line with conventional reanalyses, an idealised reanalysis would represent past states of weather for the last couple of decades as accurately as possible. In contrast to a conventional reanalysis, the goal would be to do so not with a state-of-the-art numerical weather prediction model, but instead with a model of reduced complexity.

To create these reanalyses, appropriate idealised models need to be combined with a state-of-the-art data assimilation component. Due to the reduced complexity of the model itself, the data assimilation component will be comparatively easy to implement. In fact, models of reduced complexity are sometimes used as toy models to develop and test data assimilation pro-

215 cedures[3]. Model–data assimilation pairs as they are envisioned here do thus already exist. To avoid the tremendous complexities of dealing with actual observations, I would further suggest to use existing reanalyses instead of observations.

Using this procedure, for example, based on a quasi-geostrophic model, the idealised reanalysis would represent the optimal fit between observed reality and the reduced dynamics of the quasi-geostrophic model. By design, many processes (even in the extratropics) are not represented in the quasi-geostrophic model. These processes would then appear as a forcing external to the

220 model. The forcing fields result from the data assimilation procedure either as the analysis increments, or as a parameter field explicitly included in the model formulation that is to be estimated by the assimilation procedure. In both cases, forcing fields would be available for each prognostic variable in the model, specifying the forcing required to keep the idealised reanalysis in sync with the observations or conventional reanalysis. The magnitude of this forcing relative to the tendencies due to model-internal dynamics would then indicate the degree to which the assumptions in the idealised model have been valid as a function

of time and space.

Reducing model complexity even further, a reanalysis based on a barotropic non-divergent model would constitute the optimal fit between observed reality and past Rossby wave activity. The reanalysis would exclusively represent barotropic Rossby waves, both stationary and non-stationary, their linear propagation and non-linear interactions. With such a dataset it would thus be straightforward to test the hypotheses suggested here. Further, the corresponding forcing fields constitute a

230 dataset of Rossby wave initiation and modification events that can be linked to baroclinic, diabatic and other processes by comparing with conventional and more complex idealised reanalyses. Besides testing the hypothesis suggested here, such a dataset would also be ideal to follow the suggestions of White et al. (2022) towards clarifying the relation between Rossby waves, wave guides, and extreme events.

## 6   A specific plan to test the hypotheses

Following the suggested approach would require four main steps. Step I contains the implementation of the data assimilation procedure and of the adjoint model, providing the idealised reanalysis datasets and model tools used in Steps II-IV. Step II systematically explores the relationship between transient and near-stationary waves, thus mainly addressing hypothesis 4. Step III complements Step II by linking near-stationary and transient wave activity to weather events such as cyclones, jets, and cold-air outbreaks. Step III thus directly targets hypothesis 2, and will, in combination with Step II, allow to assess the

validity of hypothesis 1. Finally, Step IV assesses the predictability of long-distance teleconnections, targeting hypothesis 3. A brief description of each step follows.

**Step I: Implementation of the data assimilation procedure and construction of the idealised reanalyses.** Suitable idealised models are readily available (e.g., Bedymo, see Spensberger et al., 2022), but will generally need to be extended by a data assimilation component. To minimise the associated risk, and to have preliminary versions of the idealised reanalyses avail-

---

[3]The Joint Effort for Data assimilation Integration (JEDI) software package contains a quasi-geostrophy model. Documentation is available under https://jointcenterforsatellitedataassimilation-jedi-docs.readthedocs-hosted.com/en/7.0.0/inside/jedi-components/oops/toy-models/qg.html, last accessed 9 February 2024.

able early-on, the data assimilation procedure can be bootstrapped, starting from spectral nudging. From a data-assimilation perspective and with the "observations" being another reanalysis, nudging is equivalent to a 3D-variational data assimilation (3D-Var) with the assumption that the error covariances (statistical representations of the model dynamics) of the two models are identical. 3D-Var is thus a natural extension of nudging, accounting for differing error covariances between the input reanalyses and idealised model. Guidance on how to construct these error covariance matrices is available from the numerical weather prediction community. With 3D-Var, each state in isolation is consistent between the input and idealised reanalyses, but differences in the evolution of the underlying models are ignored. Taking these into account, we arrive at 4D-variational data assimilation (4D-Var), the final step in the bootstrapping sequence.

For 4D-Var as well as for Step IV in this plan, an adjoint for the idealised model is required. An adjoint model complements a given model by tracing backwards in time how a particular state came to be, rather than predicting forward in time how this state is evolving (Errico, 1997). The adjoint can either be constructed through algorithmic differentiation (using for example TAPENADE, Hascoët and Pascual, 2012), or through manual derivation and implementation of the adjoint equations. Irrespective of this choice, the derivation of the adjoint model will be more straightforward than for most other atmospheric models because no parametrisations are required beyond linear relaxation and bi-harmonic diffusion. For all remaining terms in idealised model equations, well-tried standard recipes to derive the adjoint are available and can be followed.

**Step II: From transient to stationary waves.** The Rossby wave-only reanalysis produced in Step I provides the foundation for the two main analyses in Step II. First, this reanalysis can be used to decompose the Rossby wave vorticity budget into a near-stationary and a transient component following Cai and Van Den Dool (1994) and Feldstein (1998). If the traditional stationary wave perspective is correct, the near-stationary component should evolve largely independently from the transient component. If, in contrast, hypothesis 4 is correct, one would find scale interactions to be an important contributor to the near-stationary circulation. Second, one could use the Rossby-wave only reanalysis to diagnose vorticity transports due to Rossby waves. Like wave-activity fluxes, the vorticity transport highlights dynamical connections between regions, but its calculation does not require physical assumptions. The time scales on which the Tropical Indo-Pacific and the North Atlantic are connected can then simply be derived by decomposing the transport into different frequency bands.

**Step III: From chaotic weather to linear waves.** Hypotheses 1 and 2 in combination imply a synergetic relation between Rossby waves and weather events in creating teleconnection patterns over large-distances. To test the hypotheses, one could detect weather events in the reanalyses used as observations using established detection algorithms, and then relate them to Rossby wave activity, initiation, and damping using composite analyses.

**Step IV: Predictability through teleconnections.** Finally, one can apply the results from Steps II & III to the problem of predictability through long-distance teleconnections. Starting from the event to be predicted (e.g., the occurrence of a certain phase of the NAO), one can use the adjoint model to trace back predictability by identifying those processes to which the event is most sensitive (following, e.g., Galanti and Tziperman, 2003; Heimbach et al., 2011). Such processes could be scale-interactions of Rossby waves or the occurrence of a specific kind of weather event, as identified using the diagnostics of Steps II & III. Repeating the procedure using the occurrence of these key process(es) as the event to be predicted, one can thus work

one's way backwards and extract chains of processes and events that in combination yield predictability for the original event. If hypothesis 3 is correct, these key processes act predominantly on synoptic time scales.

This adjoint-based approach has two main benefits over current approaches studying the effect of a precursor event using either large ensembles or a dedicated model experiment in which the atmosphere is nudged to the desired precursor state. First, by tracing model sensitivities backwards in time, no a-priori hypothesis is necessary about which potentially important precursor events to consider. The adjoint will simply point to locations and variables to which the predictand is most sensitive, thus suggesting events or processes to be investigated further. Second, being based on an idealised model, the adjoint is orders of magnitude less demanding computationally than running a dedicated model experiment or analysing a large ensemble.

## 7    Further applications of the idealised reanalyses

The idealised reanalyses created following the overall approach have many potential applications beyond the one outlined here. For example, the idealised model Bedymo (Spensberger et al., 2022) could easily be configured to be a dry 3-layer primitive equation model. The corresponding idealised reanalysis would then be ideally suited to assess the influence of diabatic effects on the storm track. This allows one to directly address the long-standing question on their role for mid-latitude dynamics and predictability. Expanding on this idea, a hierarchy of idealised reanalyses accompanying each step in the Held (2005) hierarchy of models would simplify considerably the search for a minimal model required to represent a phenomenon of interest because its representation could then simply be compared across existing datasets without requiring a new set of model simulations.

*Author contributions.*  n/a (single-author submission).

*Competing interests.*  I declare no competing interests.

*Acknowledgements.*  This idea is based on a research proposal submitted for an ERC Starting Grant 2021 and 2022, which is in turn the result of discussions with many colleagues over several years. Of these colleagues I want to specifically thank Thomas Spengler and Madlen Kimmritz as they had a particularly large influence in shaping the ideas expressed herein. Nevertheless, all mistakes and misguided hypothesis are mine, not theirs. Finally, I want to thank the two reviewers, Volkmar Wirth and Daniela Domeisen, for their constructive comments which have helped clarify and sharpen the arguments.

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
