# Peer review of "WCD Ideas: Teleconnections through weather rather than stationary waves"

_EGUsphere, 2023_

## Referee Comment (RC1)

**Review of the manuscript** "WCD Ideas: Teleconnections through weather rather than stationary waves" by C. Spensberger, submitted for publication to Weather and Climate Dynamics

**Recommendation:** Important revisions

**General**

This manuscript challenges the traditional idea that long-distance teleconnections can fundamentally be understood in terms of long-lived quasi-stationary Rossby wave trains; rather, the author suggests that these teleconnections should be re-interpreted as due to a chain of transient nonlinear events on the synoptic scale, i.e., in terms of weather features. Part of the question is whether or not the synoptic-scale variability is too strong to allow a meaningful analysis in terms of a mean background state plus (linear) stationary wave theory.

In my eyes this manuscript is a welcome contribution to the discussion, raising issues with a long-held perspective and, at the same time, suggesting an avenue towards further progress. In some places the text appears quite opinionated with sharp formulations, while the author sketches supporting arguments. Some of the arguments may not stand the test of time, and in my text below I indicate a few points where I am skeptical. To be fair, the author does not exclude the possibility that his hypotheses might eventually be (partly) refuted (lines 107, 108). Despite these caveats I believe that the proposed hypotheses are interesting and relevant, and it is hoped that a publication in the present format will trigger further discussion.

Below I raise a few major issues. Regarding two of them (items 1 and 2) I have a rather strong opinion and I urge the author to revise the text accordingly. Regarding the remaining issues, I am aware that this idea-format differs from a regular paper: rather than presenting full analyses and final results, it is meant to talk about ideas and thereby instigate discussion. The question is to what extent the author wants to anticipate some of the discussion in his text.

**Major issues**

1. It is important to be fair regarding earlier work — especially in this new format of "WCD ideas". Therefore, all relevant previous papers need to be quoted, papers that have discussed or at least mentioned similar ideas, or possibly even the same idea.

   I think that the current text is deficient in this respect. Strictly speaking, the basic idea presented in this manuscript is not new. One important previous paper that quite explicitly discussed this idea in the context of an observed episode is that by Davies (2015). And there are probably further papers that may have touched upon this or similar ideas (e.g., Moore *et al.*, 2010), although perhaps in a somewhat implicit fashion. These earlier references

should be mentioned and briefly discussed. Also, a statement such as "no attempt has been made yet to interpret and understand long-distance teleconnections in terms of variations of weather" (lines 25, 26) seems unfair.

Similarly, doesn't this text beg for mentioning the recently suggested concept of "recurrent Rossby wave packets" (Röthlisberger *et al.*, 2019; Ali *et al.*, 2021)? In these papers the persistence of dry and wet spells is interpreted as the recurrent occurrence of Rossby wave packets, with the underlying idea being quite analogous to the idea presented in the current manuscript. In both cases a longer-term (seasonal or sub-seasonal) phenomenon is re-interpreted as the reoccurrence of a synoptic-scale weather-like phenomenon that "happens" to add up and result in the observed seasonal anomaly.

2. I have an issue with the discussion of causality (in particular the paragraph around line 70). The author seems to imply that stationary wave theory is fundamentally unable to establish cause-and-effect relationships. I would argue with this point of view. To be sure, I am aware that it is difficult to infer causality from a diagnostic relation that does not contain an explicit time derivative (something which, of course, has been known for a long time, e.g., Lorenz 1967). Geostrophic balance is but one particular example. However, I would argue that it is possible to talk about cause and effect in a forced-dissipative system in which the forcing can be considered as sufficiently "external", i.e., independent of the response. A prime example of such a situation is the theory of stationary planetary waves. To the extent that the forcing (e.g., orography) is independent of the response (the Rossby wave train emanating from the orography), I would argue that the orography can and should be considered as causing the Rossby wave train. Therefore, I disagree with the author that stationary wave theory per se is unable to infer causality. In my eyes, the issue is not the lack of causality in the original argument of Hoskins and Karoly (1981), but rather the question of why and how such a linear theory can (seemingly) be applied to long-range teleconnections in the light of strong weather systems (i.e., strong non-stationary non-linearities). The latter issue is explicitly formulated by the author in lines 63-64.

Let me be somewhat more fundamental and philosophical. To be sure, the succession of several synoptic-scale events can provide a causal chain. However, an "explanation" in terms of a complete physical chain of individual weather events (lines 98-100), although certainly being causal, may turn out to be pretty useless regarding a deeper "understanding". Extending this general approach to its extreme would mean that, for instance, the atmospheric circulation (discounting, for the sake of the argument, moist physics and further complications) can be "explained" by the validity of Newton's second law. In this perspective one simply states that each air parcel in the Earth's atmosphere is accelerated in proportion to all forces acting on the parcel divided by its mass. This statement is both true and causal! But would it help to "understand" important dynamical phenomena such as baroclinic instability? In my eyes a theory like the Eady or Charney model are more "useful" in this respect. Couldn't a similar argument apply to the issue of long-range teleconnections? I would argue that linear wave

theory, if applicable, does have some explanatory power in this case. To say it again: The crucial question is to find out why the Hoskins-Karoly theory is (seemingly) so successful.

3. The second part of the manuscript describes an avenue toward further progress: in an attempt to diagnose Rossby wave activity, the author suggests to perform some kind of data assimilation into an idealized model that is able to represent Rossby waves only. This part is rather sketchy, fair enough, and for me a number of open questions remained. However, what's more important, could it be that the author becomes subject to his own criticism? Projecting the complex reality onto a reduced dynamical model is interesting, but the results inherit some statistical nature. Possibly this prevents a solid causal interpretation, because all the processes that are not captured by the idealized model are missing in the alleged causal chain so produced.

4. There may be other avenues to increase our understanding of teleconnections in the presence of large-amplitude eddies. For instance, there have been recent developments of a theory for wave mean flow interaction that is valid for finite-amplitude eddies, such as Nakamura and Solomon (2010) or Methven and Berrisford (2015). As we argued in a recent opinion paper (White *et al.*, 2022), such new concepts may prove useful in the current context: they have the potential to push wave theory towards applications where finite-amplitude eddies play a major role, and that is exactly at issue in the teleconnection problem.

**Minor issues, typos etc.**

1. Line 39, "small changes in the flow structure. . . ": do you mean: small changes in the background state / background flow?

2. Line 50, better "a first step. . . "

3. Line 51, "North Atlantic strom track is. . . ."

4. Line 79: This section seems to suggest that linear wave theory is not a sound theory as opposed to, for instance, quasi-geostrophic theory. I do not agree. The problem is not that linear wave theory lacks foundation; rather, the problem is that it is unclear whether or why linear wave theory is applicable to the problem of teleconnections.

5. Line 86, what is a "linear mean state"? Do you mean a basic state used in the framework of linear theory?

6. Line 144, 197, "allows one to . . . ."

7. Line 183: Typo (reanalyses)

Mainz, 27 November 2023

Volkmar Wirth

**References**

Ali, S. M., O. Martius, and M. Röthlisberger 2021. Recurrent rossby wave packets modulate the persistence of dry and wet spells across the globe. *Geophys. Res. Lett.* **48**, https://doi.org/10.1029/2020GL091452.

Davies, H. C. 2015. Weather chains during the 2013/2014 winter and their significance for seasonal prediction. *Nature Geoscience*, DOI:10.1038/NGEO2561.

Lorenz, E. N. 1967. *The Nature and Theory of the General Circulation of the Atmosphere*, Volume WMO-No. 218.TP.115. World Meteorological Organisation.

Methven, J., and P. Berrisford 2015. The slowly evolving background state of the atmosphere. *Quart. J. Roy. Met. Soc.* **141**, 2237–2258.

Moore, R. W., O. Martius, and T. Spengler 2010. The modulation of the subtropical and extratropical atmosphere in the Pacific basin in response to the Madden–Julian oscillation. *Mon. Wea. Rev.* **138**, 2761–2779, doi:10.1175/2010MWR3194.1.

Nakamura, N., and A. Solomon 2010. Finite-amplitude wave activity and mean flow adjustments in the atmospheric general circulation. Part I: Quasigeostrophic theory and analysis. *J. Atmos. Sci. 67*(12), 3967–3983.

Röthlisberger, M., L. Frossard, L. F. Bosart, D. Keyser, and O. Martius 2019. Recurrent synoptic-scale Rossby wave patterns and their effect on the persistence of cold and hot spells. *J. Climate* **32**, 3207–3226, 10.1175/JCLI–D–18–0664.1.

White, R. H., K. Kornhuber, O. Martius, and V. Wirth 2022. From atmospheric waves to heatwaves: A waveguide perspective for understanding and predicting concurrent, persistent and extreme extratropical weather. *Bull. Am. Meteorol. Soc.*, https://doi.org/10.1175/BAMS–D–21–0170.1.

---

## Author Comment (AC1)

**Response to reviewers – "WCD Ideas: Teleconnections through weather rather than stationary waves"**

C. Spensberger

22 January 2024

I sincerely thank the two reviewers, Volkmar Wirth and Daniela Domeisen, for their constructive, complementary, and detailed comments to the ideas put forward in this submission. My point-by-point response to the comments appears below in blue.

**Reviewer 1 – Volkmar Wirth**

**General comment**

This manuscript challenges the traditional idea that long-distance teleconnections can fundamentally be understood in terms of long-lived quasi-stationary Rossby wave trains; rather, the author suggests that these teleconnections should be re-interpreted as due to a chain of transient nonlinear events on the synoptic scale, i.e., in terms of weather features. Part of the question is whether or not the synoptic-scale variability is too strong to allow a meaningful analysis in terms of a mean background state plus (linear) stationary wave theory.

In my eyes this manuscript is a welcome contribution to the discussion, raising issues with a long-held perspective and, at the same time, suggesting an avenue towards further progress. In some places the text appears quite opinionated with sharp formulations, while the author sketches supporting arguments. Some of the arguments may not stand the test of time, and in my text below I indicate a few points where I am skeptical. To be fair, the author does not exclude the possibility that his hypotheses might eventually be (partly) refuted (lines 107, 108). Despite these caveats I believe that the proposed hypotheses are interesting and relevant, and it is hoped that a publication in the present format will trigger further discussion.

Below I raise a few major issues. Regarding two of them (items 1 and 2) I have a rather strong opinion and I urge the author to revise the text accordingly. Regarding the remaining issues, I am aware that this idea-format differs from a regular paper: rather than presenting full analyses and final results, it is meant to talk about ideas and thereby instigate discussion. The question is to what extent the author wants to anticipate some of the discussion in his text.

Many thanks again for your detailed criticism! I am happy to hear that you found this to be a welcome contribution to instigate discussion, because this was (with good margin) my primary aim with this submission. Along the same line, I want to emphasize that my aim with suggesting these hypotheses was not be correct, but much rather to pose research questions that I think would yield relevant and insightful results irrespective of the actual outcome.

**Major issues**

**(1)** It is important to be fair regarding earlier work — especially in this new format of "WCD ideas". Therefore, all relevant previous papers need to be quoted, papers that have discussed or at least mentioned similar ideas, or possibly even the same idea.

I think that the current text is deficient in this respect. Strictly speaking, the basic idea presented in this manuscript is not new. One important previous paper that quite explicitly discussed this idea in the context of an observed episode is that by Davies (2015). And there are probably further papers that may have touched upon this or similar ideas (e.g., Moore et al., 2010), although perhaps in a somewhat implicit fashion. These earlier references should be mentioned and briefly discussed. Also, a statement

such as "no attempt has been made yet to interpret and understand long-distance teleconnections in terms of variations of weather" (lines 25, 26) seems unfair.

Similarly, doesn't this text beg for mentioning the recently suggested concept of "recurrent Rossby wave packets" (Röthlisberger et al., 2019; Ali et al., 2021)? In these papers the persistence of dry and wet spells is interpreted as the recurrent occurrence of Rossby wave packets, with the underlying idea being quite analogous to the idea presented in the current manuscript. In both cases a longer-term (seasonal or sub-seasonal) phenomenon is re-interpreted as the recurrence of a synoptic-scale weather-like phenomenon that "happens" to add up and result in the observed seasonal anomaly.

Thanks for pointing me to these references, I agree their omission is a lack with the original submission, and one that will be rectified. I am aware of these works, and they did not make it in the original submission because of different reasons.

First, regarding Moore et al. (2010) and works along similar lines by, for example, Sebastian Schemm. They are absolutely similar in basic approach, but they each focus on individual synoptic phenomena in specific regions (e.g., blocking in the Pacific Northwest, cyclogenesis in the Gulf Stream region) and link these to the tropical Pacific. Their omission in the original submission is simply due to the text being adapted from another that was subject to severe length constraints. Scientifically, what I find missing with these works is a framework that was able to link these individual spotlights into some more coherent whole, and what I propose here might ideally develop into just such a framework. It is the lack of a framework that I tried to express with the criticism you quote in L25-26. I was always thinking but apparently never wrote down a "systematic" in this sentence. With this addition, my criticism might already become more fair and understandable, but I will extend the surrounding discussion following my response here, and thus also rephrase my criticism in more detail/less broad strokes.

Second, regarding the work on recurring Rossby wave packets that in my eyes managed to systematise the case-based findings of Davies (2015). I thank specifically for pointing me to these works, as I did not fully realise the underlying analogy in approach. I agree, this line of work is very much along the lines that I propose here, except in one fundamental aspect: the focus on recurring patterns. Following your comment, I would regard my suggestions here as an extension of their arguments to those monthly and seasonal averages that are not linked to a recurring Rossby wave pattern, which, I think, should be the case for the majority of the more average seasons and months. Phrasing it like this, the analogy might seem quite obvious, but I still think there is still a bit of a conceptual leap here (and therein likely the reason why I previously did not consider them properly): for recurring patterns there is a clear one-to-one correspondence between the time mean and (selected) instances within the respective time period. I don't think that close correspondence is necessary for the same arguments to remain valid, because, I would argue, the time average remains the superposition of several transitory Rossby wave packets even without recurrence. Each packet has their distinct dynamical origin and region of dissipation that can be understood very much in parallel to how cited authors interpret their respective one recurring pattern. Following this thought, one could argue that these studies actually showed (or at least made very plausible) my hypotheses 2 and 4 for the subset of (typically more extreme) months and seasons where recurring patterns dominated.

**(2)** I have an issue with the discussion of causality (in particular the paragraph around line 70). The author seems to imply that stationary wave theory is fundamentally unable to establish cause-and-effect relationships. I would argue with this point of view. To be sure, I am aware that it is difficult to infer causality from a diagnostic relation that does not contain an explicit time derivative (something which, of course, has been known for a long time, e.g., Lorenz 1967). Geostrophic balance is but one particular example. However, I would argue that it is possible to talk about cause and effect in a forced-dissipative system in which the forcing can be considered as sufficiently "external", i.e., independent of the response. A prime example of such a situation is the theory of stationary planetary waves. To the extent that the forcing (e.g., orography) is independent of the response (the Rossby wave train emanating from the orography), I would argue that the orography can and should be considered as causing the Rossby wave train. Therefore, I disagree with the author that stationary wave theory per se is unable to infer causality. In my eyes, the issue is not the lack of causality in the original argument of Hoskins and Karoly (1981), but rather the question of why and how such a linear theory can (seemingly) be applied to long-range teleconnections in the light of strong weather systems (i.e., strong non-stationary non-linearities). The latter issue is explicitly formulated by the author in lines 63-64.

Let me be somewhat more fundamental and philosophical. To be sure, the succession of several synoptic-scale events can provide a causal chain. However, an "explanation" in terms of a complete physical chain of individual weather events (lines 98-100), although certainly being causal, may turn out to be pretty useless regarding a deeper "understanding". Extending this general approach to its extreme

would mean that, for instance, the atmospheric circulation (discounting, for the sake of the argument, moist physics and further complications) can be "explained" by the validity of Newton's second law. In this perspective one simply states that each air parcel in the Earth's atmosphere is accelerated in proportion to all forces acting on the parcel divided by its mass. This statement is both true and causal! But would it help to "understand" important dynamical phenomena such as baroclinic instability? In my eyes a theory like the Eady or Charney model are more "useful" in this respect. Couldn't a similar argument apply to the issue of long-range teleconnections? I would argue that linear wave theory, if applicable, does have some explanatory power in this case. To say it again: The crucial question is to find out why the Hoskins-Karoly theory is (seemingly) so successful.

Sincere thanks for opening the discussion on a more fundamental level, I very much enjoy the opportunity to discuss these issues further here.

First, regarding your arguments about causality with stationary waves. I actually largely agree with your arguments for the specific example that you focused on—stationary waves excited by topography. Here, I agree, the forcing can be considered external, and I would add that the forcing is also unquestionably stationary. Both of these assumptions I find hard to justify for tropical convection, variations of which cause the teleconnections in question. Tropical convection pulses diurnally and in case of the MJO it propagates over thousands of kilometers within a time scale of pentads, which for me makes it hard to regard tropical convection as a stationary forcing. Further, many studies have documented a mid-latitude influence on tropical convection (sec. 2.2 in Stan et al., 2017, and references therein), not the least a link between the NAO and MJO initiation events (Lin et al., 2009), which to me indicates that the forcing cannot be considered external either.

Second, regarding the discussion about what constitutes an explanation and understanding. I am again very much with you in that, in a hierarchy of explanations, a more fundamental one is not necessarily more useful for conceptual understanding than a more abstract one, and often actually less so. At the same time, I feel the argument is missing its target, because I am not arguing to abandon all high-level explanations and return to first principles only. Instead, I would like to point out what I perceive to be a gap in our hierarchy of explanations which is located in-between transient, chaotic, and non-linear weather on one side, and linear stationary waves on the other side. It is to make apparent this gap that I point out the limitations of stationary wave theory, not to abandon the theory entirely.

Returning to the analogy in the manuscript: you need inertia-gravity waves to explain how geostrophic balance can be attained in practice, i.e. why geostrophic balance is not only a mathematical construct but something that is more-or-less realised in nature. Similarly, we will need to go beyond stationary wave theory to explain "why the Hoskins-Karoly theory is (seemingly) so successful". I here propose first go back to weather to then construct an intermediate level of abstraction that can serve as an intermediate step linking the so-far unconnected levels of explanation. This is what I want to express by stating: "If correct, hypothesis 2 provides the missing conceptual foundation. Non-stationary finite-amplitude Rossby waves regularly propagate approximately linearly over large distances (e.g., Wirth and Eichhorn, 2014; O'Brien and Reeder, 2018). They further have a clear influence on the non-linear evolution of mid-latitude weather, for example by determining the predominant locations of cyclogenesis (Holton and Hakim, 2013). It thus seems plausible that they constitute the ordering principle that links the non-linear instantaneous weather to a linear mean state." (L82-86).

May be putting it in yet-other words: in my view stationary Rossby waves are a special case (possibly an important one) in a similar vein as mean patterns dominated by recurring Rossby wave packets are a special case of mean patterns in general. All Rossby waves all the time communicate some forcing over large distances, why should it be exclusively stationary waves that create long-distance teleconnections? In particular if the forcing is only partly external, and clearly not stationary either?

With these supplemental explanations, I trust it becomes apparent that I do not intend to portrait linear-wave theory as unsound (cf. minor comment on L79).

**(3)** The second part of the manuscript describes an avenue toward further progress: in an attempt to diagnose Rossby wave activity, the author suggests to perform some kind of data assimilation into an idealized model that is able to represent Rossby waves only. This part is rather sketchy, fair enough, and for me a number of open questions remained. However, what's more important, could it be that the author becomes subject to his own criticism? Projecting the complex reality onto a reduced dynamical model is interesting, but the results inherit some statistical nature. Possibly this prevents a solid causal interpretation, because all the processes that are not captured by the idealized model are missing in the alleged causal chain so produced.

This criticism seems a bit unfair to me. The current state of affairs is that many-a wave pattern observed

in a time mean is subjectively interpreted as a stationary wave, not infrequently without ever verifying the interpretation with an actual stationary wave model. And even in those cases where a comparison is made, the degree of similarity between the linear model and the observations/realistic model remains a matter of subjective judgement.

In contrast, I propose here

1. to use a rigorously defined and well-tested mathematical procedure to map results from more realistic models to more idealised ones; a procedure which in addition provides a measure that quantifies how well the two models fit together at every point in time and space.

2. to vastly increase the complexity of the idealised model by which one is aiming to capture the essential dynamics of the problem, moving from a linear stationary wave model to (first) a fully non-linear barotropic model and (second) a quasi-geostrophic model.

It is of course possible that, despite the increased complexity, there still is essential dynamics that is not captured in the idealised models. Technically this would become apparent through the suggested approach in that the idealised models require large amounts of forcing at most places and times to keep in sync with the more realistic model. But if this was the case, this would also cast doubt on all studies trying to make sense of mid-latitude dynamics in terms of propagating Rossby waves and quasi-geostrophy. While not strictly impossible, that seems very implausible to me. And besides: were there really a serious and systematic problem with these bed-rock fundamental concepts of mid-latitude dynamics, it'd be better to find out rather sooner than later!

More realistically, there will be more-or-less isolated instances in space and time where the idealised models cannot represent what is happening in the more realistic model, and thus locally and temporarily require large amounts of forcing to stay in sync. This would not invalidate the outlined approach, but constitute an essential basis for the analyses that I propose here. Using for example the barotropic reanalysis, the forcing dataset would point at locations in space and time where Rossby waves are initiated, modified or dissipated by processes external to the barotropic model. Such a dataset would in my eyes be extremely helpful, both to address the questions I raise in this manuscript, but also for the wider community. Also for other idealised representations of the atmosphere it seems extremely helpful to have available datasets quantifying how valid the different assumptions have been at every point in space and time over the reanalysis history.

I assume this comment mainly reflects that the overall approach that I suggest in the manuscript has not become fully clear. I will thus expand this section in the revised submission, in particular to better explain the ideas behind and potential uses of the idealised reanalyses.

**(4)** There may be other avenues to increase our understanding of teleconnections in the presence of large-amplitude eddies. For instance, there have been recent developments of a theory for wave mean flow interaction that is valid for finite-amplitude eddies, such as Nakamura and Solomon (2010) or Methven and Berrisford (2015). As we argued in a recent opinion paper (White et al., 2022), such new concepts may prove useful in the current context: they have the potential to push wave theory towards applications where finite-amplitude eddies play a major role, and that is exactly at issue in the teleconnection problem.

I agree, and will include/expand the discussion of these works and how they relate to the approach I am suggesting in the revised submission.

**Minor issues**

**Line 39** "small changes in the flow structure. . . ": do you mean: small changes in the background state / background flow?

Yes, that is what I meant, and I will reword accordingly.

**Line 79** This section seems to suggest that linear wave theory is not a sound theory as opposed to, for instance, quasi-geostrophic theory. I do not agree. The problem is not that linear wave theory lacks foundation; rather, the problem is that it is unclear whether or why linear wave theory is applicable to the problem of teleconnections.

I do not mean to imply that linear wave theory is generally unsound. Specifically to avoid this misunderstanding, I write only a few lines earlier (L74): "Following the analogy, both geostrophy and stationary wave theory are indispensable fundamental concepts for understanding mid-latitude flow." I thus hope this specific issue is resolved by my response to major issue 2.

**Line 86** what is a "linear mean state"? Do you mean a basic state used in the framework of linear theory?

Thanks for pointing out the unclear wording. I meant to refer to "time-mean states that follow expectations from linear (stationary) theory." I will reword along these lines.

**Typos, etc.**

Thanks for pointing out these typos and mistakes, they will be fixed.

**Reviewer 2 – Daniela Domeisen**

**Major comments**

**(1)** In my opinion the use of the term "teleconnection" for quasi-stationary patterns such as e.g. the North Atlantic Oscillation is incorrect. The term should be used for what was originally defined in Wallace & Gutzler 1981 (also cited in this study), which defines teleconnections as correlations between time variations in meteorological parameters at separated points on the globe. Note that this is my personal opinion, and I am aware that patterns such as the NAO are often referred to as "teleconnection patterns". However, the views of "local" and "global" teleconnections sometimes appear mixed in the manuscript, and this should be clarified (see detailed comments below).

Thanks for the comment. For me the need to distinguish between local and long-distance teleconnections only became obvious when writing this manuscript, so it's good to read that I am not alone in seeing the need to clearly separate these two concepts.

I am however less sure about a clear historical definition by Wallace and Grutzler (1981). Admittedly, they require teleconnections to be "between [. . . ] widely separated points on earth" in their first sentence of the introduction. But already on the following page (their section 2a), they use this term to discuss different definitions of North Atlantic Oscillation, including ones that I think we both would consider to be local. I thus don't think one can claim that Wallace and Grutzer clearly separated these concepts, and, as you point out yourself, practical usage since surely has not.

Still, I want to clearly separate these concepts in this work, and I will thus make sure for every instance in which I use the term "teleconnection" that it's clear whether I refer to local or long-distance variant. I do not have a strong opinion on this issue, but I fear that reserving the term "teleconnection" to exclusively refer to the long-distance variant will be more confusing than helpful to all but very careful readers.

**(2)** Stationary wave patterns, such as e.g. the Aleutian low, are clearly made up of smaller-scale processes on shorter timescales – this is well known (e.g. Orlanski, 2005). I don't think there is much debate of this point, but of course in the sub-seasonal to seasonal literature this question is not discussed, as it is the average state or persistence of these large-scale patterns that matters for long-range teleconnections.

I agree. For what I call local teleconnections, there are often straightforward and generally accepted dynamical interpretations in terms of variations of occurrence or properties of single weather systems. I here put forward a number of arguments for why similar arguments might be true for long-distance teleconnections, but then in terms of variations in the occurrence of typical synoptic storylines (cf. your comment on L105-106) that realise an essential part of the teleconnection. Should the hypotheses I put forward generally hold, also studies of sub-seasonal to seasonal predictability would benefit from adopting the concepts I am suggesting (cf. also my response to detailed comment L101-106).

**(3)** "idealized reanalysis": This is an interesting idea. It's however not clear how such a reanalysis would look. Would e.g. other types of waves (gravity waves, Kelvin waves, mixed Rossby-gravity waves, etc) also be part of the dataset? Which variables would be part of the reanalysis?

The variables available and the dynamics represented will depend only on the specific idealised model used to create the idealised reanalysis. For example, using the model Bedymo in a QG configuration (one of the configurations I am proposing to test my hypotheses), there is in essence only one prognostic variable (temperature), which, in combination with the (prognostic) surface pressure as boundary condition, determines the entire model state via QG balance assumptions. Thus, geopotential, horizontal and vertical wind components could all be diagnostically derived, and would in practice surely also be made available as part of the reanalysis.

In addition, the reanalysis would for this case provide both temperature and surface pressure forcing fields required to keep the idealised model more-or-less in sync with the more realistic model used as observations in the data assimilation. These forcing fields will vary in space and time, pointing out places and times where (in this example) the QG model cannot represent what's happening in the more realistic model (and thus requires a lot of forcing to keep in sync), and likely periods and areas where QG does not require much forcing to represent the more realistic model result.

In terms of dynamics, there would by QG-definition not be any (mixed-)inertial waves, so in essence the model would represent baroclinic instability and Rossby waves, their non-linear interactions, and not much else.

I realise, also in light of comment 3 of reviewer 1 (Volkmar Wirth), that my ideas around the idealised reanalyses have not fully come across in the original submission. I will thus expand this section to explain the ideas behind and potential uses of the idealised reanalyses in more detail.

**(4)** Step IV: "Predictability through teleconnections.": currently, predictability and causality are generally evaluated in models using large ensembles that either contain a specific precursor event or not, and hence the causality can easily be established. Another method involved nudging of the atmosphere to a certain state to establish causality. It would be helpful to clarify what the approach through the idealized reanalysis would add to understand the relevant mechanisms.

Thanks for the question. In my view, the main advantage of using the adjoint is that the model itself will point out those locations and variables in previous time steps to which the solution in the current time step is most sensitive to. These locations will probably often match expectations from existing literature, but I am sure there will be surprises where the most sensitive region and variable have not been explored through the outlined kinds of experiments yet.

In addition it might be worth mentioning that this method would be many orders of magnitude cheaper computationally than running (or even analysing) a large ensemble or running a dedicated nudging experiment for every parameter and region that might play a role for the predictand in question.

**(5)** Step IV: what about precursor processes that are crucial for causing an event that do not involve Rossby waves? For example, tropical convection is crucial for ENSO and MJO teleconnections, but if I understand correctly they would not be represented in the idealized reanalysis.

They will be represented not in the model dynamics, but still be part of the reanalysis as the forcing required to keep the idealised model in sync (cf. response to major comment 3). By matching space and time, the more realistic model will then help pinpoint which physical processes lead to large idealised model forcing.

Carrying out this analysis in practice, I am sure there will be surprises also here, where the idealised model forcing is not in the locations at the times where one might previously have expected, and these cases will then be particularly interesting to dig into further.

**(6)** Following up on point 5, and more generally (beyond step IV described in the manuscript), I think the author may be talking not about teleconnections, but about Rossby wave trains. Teleconnections involve a much wider range of processes than just Rossby waves, including processes in both the atmosphere and the ocean (see e.g. Liu and Alexander, 2007). For the concrete example of ENSO and MJO teleconnections to the North Atlantic that is extensively used in the manuscript, this teleconnection includes a myriad of other processes beyond Rossby waves, see specific point below (lines 116 -121). Given the use of this teleconnection for illustration throughout the manuscript I would encourage the author to refer to the literature covering this teleconnection in more depth (see detailed comment below).

Thanks for bringing this up, I very much agree with the need to include this context. I had included a discussion of this issue in an extended earlier version of this manuscript and I acknowledge that this context is lacking in my original submission. I will include a discussion of how the ideas I present in this manuscript relate to the body of work on teleconnections more teleconnections more generally; based on the review you suggest as well as the more recent and targeted review of interactions between the MJO and the North Atlantic by Stan et al. (2017). With the MJO forcing clearly being non-stationary, I think the focus on the MJO will both make my arguments more convincing and keep the required discussion of different teleconnection pathways between these regions at a length reasonable within this relatively short manuscript. I still think many of my ideas would translate to ENSO-based teleconnections, but I will remove remove all references to these from the manuscript and leave this work as a potential extension should the suggested ideas and hypotheses turn out to be useful. Further, I will state clearly as a caveat that my ideas and hypotheses pertain only the stationary-wave component that constitutes a part in most long-distance teleconnections.

**Detailed minor comments**

**Line 11** "Teleconnections can be comparatively local" is not something I would agree with, see major point 1 above.

Please refer to my response to your major comment 1.

**Line 22** "Teleconnections to ENSO and the MJO": not clear, do you mean "teleconnections originating from ENSO or the MJO"?

Yes, that is what I meant. But as in response to your major comment (6), I will remove references to ENSO from the revised manuscript, including this one.

**Lines 61-62** the manuscript sometimes goes back and forth between the two definitions of "local" teleconnections and "global" teleconnections. Although the manuscript is said to focus on the global definition, sometimes it falls back to the local definition, e.g. "exchange [...] within teleconnections". See major point 1 above.

Thanks for pointing out this confusion, I will for every instance clarify in the revised submission with variant I refer to.

**Line 54** Hypothesis 3 is formulated much less general than the first two hypotheses. It might be beneficial for the paper to stay at a conceptual level.

True. The original motivation for this more specific hypothesis was to explain my thoughts based on this specific example. But I agree, the hypothesis is in principle meant to be as conceptual as the others and will thus be rephrased in that way.

**Line 73** "they are an expression of their existence": I assume "they" refers to "stationary waves" and "their" refers to "teleconnections", is this correct? Please clarify.

Yes, you interpreted correctly. Still, thanks for pointing out the potential for confusion. I will clarify.

**Line 96** "potential and limits for predictability through these teleconnections": yes, indeed. See e.g. Gonzalez-Aleman et al, 2021

Thanks for the pointer, I was not aware of that study. Despite the somewhat different context, it fits the overall theme of my hypotheses very well in that subseasonal predictability in this case study is mediated by individual weather events. I am happy to include this citation, although I may in the end decide to instead include it as another argument to make plausible my hypothesis in section 2.

**Lines 97-100** I don't agree here. Even for considering teleconnections from a global perspective, and in particular for the example used here for the MJO teleconnection to the North Atlantic, each step is usually considered in isolation, see also point below for lines 116-121 and major point 6.

I agree that my formulation was imprecise here. My approach would only replace the step(s) involving stationary Rossby waves, resolving the synoptic processes that in the time-mean would yield the wave pattern. I will rephrase accordingly.

**Lines 101-106** note that this "paradox" is only a paradox for some models. It would be helpful to understand how the approach here can "solve" the paradox.

Interesting, thanks for pointing this out. I had perceived the "paradox" as a problem shared by many-to-all seasonal prediction models, although of course to a varying degree. I will weaken the problem statement accordingly.

Further, I agree, the description of how my approach might help was quite superficial. In more detail, my reasoning is: given we have established through the suggested activities a few typical storylines (cf. comment L105-106) of how, on synoptic time scales, the signal of varying convection in the tropical Indo-Pacific is communicated to the North Atlantic. One can then for every step in the storyline use the adjoint model to point out the variables and locations to which the given step is most sensitive. This information in combination with the forcing patterns in the idealised reanalyses around those locations should be very useful to narrow down the number of potential processes causing the prediction model to be overdispersive. I will expand the discussion in this section along this line.

**Lines 105-106** could you elaborate how this is different from the storyline approach?

The approach I suggest here to understand teleconnections as chains of events is indeed similar to the storyline approach proposed by Shepherd et al. (2018). I had not thought about my approach in that way,

so thanks for bringing up the analogy! I will include a reference to their work in the revised manuscript. An important difference seems to be, though, that my approach complements a storyline-like take on teleconnections with quantitative analyses that aim to identify probabilities of occurrence and the physical conditions under which the different links in the chain can operate. This quantitative complement should be possible for teleconnections, because we there have an archive of many "teleconnection events" that can be systematically explored for shared essential characteristics.

**Lines 116-121** Rossby waves communicating effects of ENSO and the MJO to the North Atlantic have a myriad of roles. It seems from the manuscript, e.g. also lines 144-145, that the author suggests that there is a single process acting all the way from the Indo-Pacific to the North Atlantic through Rossby wave propagation that communicates the effects of ENSO and the MJO to the North Atlantic. First of all, Rossby waves are not the only process that communicates these effects, e.g. for the pathway through the tropical North Atlantic, where the Walker circulation plays a major role. For the pathways that are communicated at least in part by Rossby waves, there is a very wide range of processes. For example, for the pathway of ENSO to the North Atlantic (this pathway has been shown to be the dominant pathway), there are the following processes involved:

- Communication of SST anomalies to tropical convection

- Tropical convection to North Pacific storm track and Aleutian low (e.g. Deng, Y., and T. Jiang, 2011)

- Rossby wave propagation to upper stratosphere

- Wave-mean flow interaction from upper stratosphere to lower stratosphere

- Lower stratospheric mean temperature anomaly to tropospheric synoptic eddies in the North Atlantic

This is just an example showing how many different processes beyond Rossby waves are involved. Given the message of the manuscript, it would make sense to reduce the analysis to Rossby wave trains rather than teleconnections (see major point 6 above).

Thanks again for pointing out this omitted context and the specific sentence in L144-145. I will rephrase that sentence and more generally adapt the manuscript as outlined in my response to your major comment (6).

**Technical comments**

Thanks for pointing these out, they will be fixed.

**Additional references**

Lin, H., Brunet, G., and Derome, J.: An Observed Connection between the North Atlantic Oscillation and the Madden–Julian Oscillation, Journal of Climate, 22, 364–380, https://doi.org/10.1175/2008JCLI2515.1, 2009.

Shepherd, T. G., Boyd, E., Calel, R. A., Chapman, S. C., Dessai, S., Dima-West, I. M., Fowler, H. J., James, R., Maraun, D., Martius, O., Senior, C. A., Sobel, A. H., Stainforth, D. A., Tett, S. F. B., Trenberth, K. E., van den Hurk, B. J. J. M., Watkins, N. W., Wilby, R. L., and Zenghelis, D. A.: Storylines: an alternative approach to representing uncertainty in physical aspects of climate change, Climatic Change, 151, 555–571, https://doi.org/10.1007/s10584-018-2317-9, 2018.

Stan, C., Straus, D. M., Frederiksen, J. S., Lin, H., Maloney, E. D., and Schumacher, C.: Review of Tropical-Extratropical Teleconnections on Intraseasonal Time Scales, Reviews of Geophysics, 55, 902–937, https://doi.org/10.1002/2016RG000538, 2017.

---

## Author Response (AR2)

**Response to reviewers – "WCD Ideas: Teleconnections through weather rather than stationary waves"**

C. Spensberger

8 March 2024

I sincerely thank the two reviewers, Volkmar Wirth and Daniela Domeisen, for their re-assessment of this manuscript and the again constructive and useful comments. A point-by-point response to the comments appears below in blue.

**Reviewer 1 – Volkmar Wirth**

**General comment**

The author revised the manuscript carefully and addressed most of my concerns - except the issue about causality, which I am still not happy with (see below).

While reading the revised manuscript I found a few minor issues, and I am not sure whether these are new ones of I just did not stumble acrosse them upon my first reading. In any case I think these minor issues should be addressed before publication.

Overall I think this is thought-provoking and useful contribution which should definitely published.

I am happy to read that I could address most of the reviewer's concerns, and I hope that I will be able to resolve a few more here.

**Minor issues**

**Line 42-43** you may want to add a recent reference about the limit of (intrinsic and practical) predictability in midlatitudes, namely T. Selz, M. Riemer, and G. C. Craig: The transition from practical to intrinsic predictability of midlatitude weather. J. Atmos. Sci., 79, 2013–2030, 2022.

True, thanks for suggesting the addition. Added.

**Line 50** "They further require spatial variations in the mean state to be gentle enough to not interfere with wave propagation." Is that really true? To be sure, linear wave theory requires quadratic terms to be small compared to the linear ones, but does it make any assumptions about spatial variations? The latter sounds as if you are referring to the WKB approximation. However, the WKB approximation is not part of linear wave theory (although it is sometimes made in addition to linear wave theory).

Thanks for pointing out this mistake, I indeed had the WKB approximation in mind when formulating this sentence. I removed the sentence.

**Line 93** I still do not see how causality (in general) is lost in stationary wave theory. As I said earlier, the (external) forcing is the cause for the emerging wave train. In my eyes, this is NOT analogous to the diagnostic relation between the geostrophic wind and the pressure gradient, which clearly precludes any interpretation in terms of cause and effect.

Thanks for bringing that point up again, I think I now understand a bit better what you wanted to criticise. It is my impression that the disagreement is mainly due to the reviewer interpreting the analogy more strictly than I had intended. In a strict sense, I agree, if all conditions for stationary wave theory were met, the theory could provide a causal explanation. Only, the conditions are clearly not met in practice.

Beyond the questions of stationarity and the degree of externality of the forcing discussed earlier, may be the conceptually largest problem is that stationary wave theory implicitly assumes that there is no weather happening on top. Any cyclone development anywhere along the stationary wave would

constitute a non-linear interaction of a finite-amplitude eddy with the stationary wave, and thus clearly invalidate the assumptions underlying the theory. Returning with this thought to the question of causality, I see the problem not with the conceptual construct of stationary waves, but in the practice of applying the concept to wavy patterns observed in a time-mean.

I realise I previously used the concept of stationary wave theory and the practice of interpreting time-mean wave patterns interchangeably, thus muddying my criticism in the analogy. I now rephrased the analogy to be more precise with my criticism, i.e., such that it only applies to time-mean wave patterns. I further moved the discussion of how stationary wave theory is not really filling the gaps in our understanding of how to interpret time-mean wave pattern closer to the analogy. I hope in now becomes clear now also in my writing that I agree with the reviewer in that the problem is not the theory itself but its application.

I am repeating myself, but still: thanks for bringing this up again, allowing me to sharpen in my arguments!

**Line 118** "[. . . ] because it transforms teleconnections from statistical relations to a causal chain of events [. . . ]": well, to the extent that linear wave theory applies, it does provide a causal explanation in my eyes. Again, the problem is not that the Hoskins-Karoly theory lacks causality, the problem is rather to understand why or whether it can be applied to the problem of teleconnections.

Here I am not directly referring to Hoskins and Karoly, but only about "shifting the focus from monthly and longer time scales to synoptic time scales". I thus trust that this comment will be solved indirectly by the more precise formulation of the analogy around Line 93, which provides a more precise context for the statement here.

**Line 152** better "[. . . ] the EP-flux, the divergence of which [. . . ]"

True, thanks for the correction.

**Line 169** "[. . . ] the required mean state cannot represent zonal asymmetries [. . . ]", well, this drawback can also be overcome, at least in a practical sense, see C. Polster and V. Wirth: A new atmospheric background state to diagnose local waveguidability. GRL, 50:DOI:10.1029/2023GL106166, 2023.

Thanks for making me aware of this latest development. I am happy to mention the study here.

**Line 182** "[. . . ] the mean state (Fig. 3a, d) remains a poor representation of the varying conditions non-stationary Rossby waves might encounter [. . . ]": just to be sure, the novel background state of Polster and Wirth (2013) can be computed from just a single snapshot, and it does vary (smoothly, though) from day to day.

You make a good point in Fig 3. However, the problem may be not so much the need to choose a background state in order to define a "wave", but rather that in the past the chosen background state was inappropriate.

I agree, your approach in Polster and Wirth (2023) appears quite promising for extracting wave guides. Still, given that Polster and Wirth (2023) was published only a few weeks ago, I would claim the final verdict here is still out. In any case, good to have several avenues to pursue!

**Reviewer 2 – Daniela Domeisen**

**Comments**

**Line 26** it's not clear what "this idea" refers to here. It sounds like it refers to "The stationary wave paradigm" used in the sentence before, but I don't think that's what the author intended.

Thanks for pointing out this potential source of confusion. I rephrased using first person voice, i.e. "[. . . ] I challenge the paradigm [. . . ]" instead of "[. . . ] the idea challenges the paradigm [. . . ]".

For the parts describing the MJO teleconnection, I have a few more suggestions for references that may be useful, especially with respect to the prediction aspect, I'm listing them in the "references" section. I leave it up to the author to decide which ones (if any) are useful for this manuscript.

Thanks for pointing me to all the additional references, I read them with interest. In the end, I included Garfinkel et al. (2014) as additional context for the MJO-North Atlantic teleconnection and Vitart (2017) and Stan et al. (2022) as additional context for practical predictability through this teleconnection.

I think all the references to Figure 3 should refer to Figure 2 instead, as Figure 2 is not referred to in the text, please correct (apologies if I missed it).

Thanks for pointing out this mistake! You are absolutely right, there are only two Figures. I corrected the references.

Overall, the manuscript is still rather Europe-centric. I understand this might be the goal, in which case I do not want to interfere here, but I would like to point out that it would be rather straight forward to make the manuscript more globally applicable by considering e.g. MJO teleconnections to other extratropical regions, such as e.g. line 46: the MJO is important in S2S prediction not just for the North Atlantic region.

I agree with the observation and I agree that it's a caveat of the manuscript as it is organised. I still prefer to keep the Euro-centrism. In the same way as the narrowing from tropical Indo-Pacific to only and specifically the MJO has helped clarify the arguments, I fear that opening to MJO teleconnections in other regions would make the arguments less clear again. Considering, for example, a hypothetical jet streak over the North Pacific which is related to strong tropical convection in the tropical Pacific. For such a case, I would not even be sure if I wanted to apply the label "long-distance teleconnection" as the connection between the regions can (in this hypothetical case) be explained solely by a variation of a single dynamical entity, that is a locally and temporarily amplified Hadley circulation.

**Lines 46-51** I think it needs to be made clear here that the first sentence talks about deterministic predictability, while S2S prediction, which is the focus of this study, is entirely based on ensemble prediction. I would recommend to avoid mixing the two concepts here. We considered the deterministic limit here, maybe interesting: Domeisen et al: "How predictable are the Arctic and North Atlantic Oscillations? Exploring the variability and predictability of the Northern Hemisphere." Journal of Climate 31.3 (2018): 997-1014.

Once more, thanks for pointing out this potential for confusion. I now make it clear that I initially only refer to deterministic probability. The transition from deterministic to ensemble predictability was already explicitly marked in the text. Thanks also for pointing me to this study; I agree it provides interesting context here and included it in the reference list discussing deterministic potential predictability.

**Line 93** hence, a "teleconnection event" would be a case where the MJO has a teleconnection to the North Atlantic

Yes. I reformulated to emphasize this aspect a bit more.

**Section 6** I appreciate the additions in this section which helped a lot to clarify the plans / concept of this study.

Good to read, thanks for the positive feedback!

**Technical comments**

Thanks for pointing these additional mistakes, they are fixed.